# Free and Bioavailable Vitamin D Are Correlated with Disease Severity in Acute Pancreatitis: A Single-Center, Prospective Study

**DOI:** 10.3390/ijms26125695

**Published:** 2025-06-13

**Authors:** Darko Siuka, Matej Rakuša, Aleš Vodenik, Lana Vodnik, Borut Štabuc, David Štubljar, David Drobne, Aleš Jerin, Helena Matelič, Joško Osredkar

**Affiliations:** 1Department of Gastroenterology, University Medical Centre Ljubljana, 1000 Ljubljana, Slovenia; darko_siuka@yahoo.com (D.S.); ales.vodenik@kclj.si (A.V.); lana.vodnik@kclj.si (L.V.); borut.stabuc@kclj.si (B.Š.); david.drobne@kclj.si (D.D.); 2Faculty of Medicine, University of Ljubljana, 1000 Ljubljana, Slovenia; matej.rakusa@gmail.com; 3Department of Endocrinology, Diabetes and Metabolic Disease, University Medical Centre Ljubljana, 1000 Ljubljana, Slovenia; 4Department of Research and Development, In-Medico, 8330 Metlika, Slovenia; 5Institute of Clinical Chemistry and Biochemistry, University Medical Centre Ljubljana, 1525 Ljubljana, Slovenia; ales.jerin@kclj.si (A.J.); helena.matelic@gmail.com (H.M.); 6Faculty of Pharmacy, University of Ljubljana, 1000 Ljubljana, Slovenia

**Keywords:** vitamin D deficiency, acute pancreatitis severity, free and bioavailable vitamin D, inflammation and immune response, clinical outcomes

## Abstract

Acute pancreatitis (AP) is primarily caused by inflammation and immunological responses, both of which are regulated by vitamin D. The purpose of this study was to examine the correlation between the severity of AP and vitamin D levels, including its total, free, and bioavailable forms. Eighty individuals with AP were enrolled in this study. Serum levels of free 25(OH)D_3_, bioavailable 25(OH)D_3_, and total 25-hydroxyvitamin D 25(OH)D_3_ were assessed. The severity of the disease course was assessed by scoring systems (Revised Atlanta classification, Ranson score, CTSI). Vitamin D deficiency was common in AP patients, with 31.3% being categorized as deficient (<50 nmol/L) and 27.5% having a severe deficiency (<30 nmol/L). Compared to patients with adequate vitamin D status, those with lower vitamin D levels had a significantly higher risk of developing moderate-to-severe AP (44.7% vs. 14.3%, *p* = 0.029). Patients with severe vitamin D insufficiency were the only ones who experienced severe AP. Clinical outcomes showed similar correlations: patients with significant vitamin D deficiency had longer hospital stays (mean of 12.1 ± 5.3 days vs. 7.8 ± 3.4 days, *p* = 0.018) and higher rates of ICU admission (31.8% vs. 8.0%, *p* = 0.007). Low levels of total, free, and bioavailable vitamin D were significantly associated with the severity of AP and ICU admission. Free, bioavailable, and total vitamin D were correlated with the severity of acute pancreatitis. All severe cases occurred in patients with severe vitamin D deficiency. Given the observational design, these associations require confirmation in interventional or mechanistic studies.

## 1. Introduction

Vitamin D deficiency poses a serious public health issue, especially in areas with limited sunlight. According to studies, a substantial portion of the population has below-optimal vitamin D levels, particularly in the winter and spring [1]. Due to reduced solar exposure, changed levels of albumin and vitamin D-binding protein (VDBP), and malabsorption, patients with gastrointestinal diseases are particularly susceptible to vitamin D insufficiency and deficiency [2,3].

Vitamin D is a fat-soluble secosteroid that helps maintain healthy bones by controlling the metabolism of calcium and phosphate. When exposed to ultraviolet B (UVB), it is produced in the skin; alternatively, it can be acquired through the diet. The primary circulating form of vitamin D, 25-hydroxyvitamin D (25(OH)D_3_), is formed in the liver through hydroxylation of vitamin D. Further hydroxylation in the kidneys creates the biologically active form, 1,25-dihydroxyvitamin D (1,25(OH)_2_D_3_), which oversees calcium absorption, immune responses, and cell differentiation. Vitamin D deficiency has been associated with a range of diseases, including osteoporosis, cardiovascular issues, and specific cancers, as well as autoimmune and inflammatory disorders. Lowered quality of life, higher morbidity and mortality, sarcopenia, and infections are also linked to this deficit [3,4,5,6,7].

Acute pancreatitis (AP) is a severe inflammatory condition affecting the pancreas that can result in systemic complications and organ failure. Vitamin D might play a complex but crucial role in AP. Vitamin D may influence the severity of AP, with an impact on inflammation, immunological responses, and calcium metabolism. According to research, by altering the production of pro-inflammatory cytokines, vitamin D regulates both innate and adaptive immune responses [8]. Vitamin D has an anti-inflammatory impact on macrophages, with increasing interleukin (IL)-10 and decreasing inflammatory stimuli (i.e., IL-1β, IL-6, tumor necrosis factor-α (TNF-α), receptor activator of nuclear factor kappa-Β ligand, and cyclo-oxygenase-2 (COX-2)). 1,25-(OH)_2_D_3_ also has an anti-oxidative effect on monocytes by upregulating glutathione reductase (GR), which results in the reduced formation of oxygen radicals [9]. With this mechanism, vitamin D might mitigate the cytokine storm in AP by controlling the expression of inflammatory cytokines that are known to be elevated in AP [10]. Studies showed the correlation between vitamin D deficiency and the exacerbation of the inflammatory response in individuals with AP, resulting in poorer clinical outcomes [11,12,13,14].

Malheiro et al. showed a correlation between cytokine levels in AP and the severity of AP [15]. AP patients showed increased levels of interleukin (IL)-6, IL-8, IL-10, vascular endothelial growth factor (VEGF), tumor necrosis factor (TNF)-alpha, and monocyte chemoattractant protein (MCP)-1 at admission when compared with healthy controls. Pancreatic necrosis and systemic effects of AP in vitamin D-deficient circumstances may be caused by excessive inflammation and immunological dysregulation [11,14,16,17]. Additionally, vitamin D supports gut barrier integrity. A deficiency in intestinal permeability can lead to bacterial translocation, which can exacerbate the severity of AP [18]. Moreover, vitamin D is necessary for calcium homeostasis, and dysregulated calcium signaling may increase the activation of pancreatic enzymes, thus intensifying pancreatic inflammation [19,20].

Recent studies emphasize how crucial it is to assess vitamin D’s free and bioavailable forms in addition to its overall amount. The free hormone theory states that only the unbound form of vitamin D may enter cells and have an impact on biology. Measuring free and bioavailable vitamin D offers a more accurate assessment of functional vitamin D status in conditions like AP, where VDBP and albumin concentrations may be changed [6,13,14]. A study by Gallerani et al. showed the existence of circannual variation in the onset of AP, with a significantly higher frequency of events in the spring and with higher mortality from December to February [21]. Studies have also shown that levels of both bioavailable vitamin D and free vitamin D vary seasonally, with higher levels generally observed in the summer months and lower levels observed in the winter months [22,23].

By assessing serum levels of total, free, and bioavailable vitamin D, our study seeks to determine the association between vitamin D status and the severity of acute pancreatitis. In order to ascertain if vitamin D levels can function as independent predictors of the course of AP, we additionally investigate their correlation with clinical outcomes, such as length of hospital stay, ICU admission, and disease severity.

## 2. Results

The analysis comprised eighty AP patients in all. The basic characteristics of these patients are presented in Table 1. The sex distribution was about equal (56.3% male), and the mean age was 58.8 ± 16.1 years. The majority of patients were alcohol consumers; however, only 19 (23.8%) cases had alcoholic AP. Biliary type was present in 38 (47.5%) patients. Overall, 30 patients (37,5%) were smokers.

Vitamin D deficiency was very common: 31.3% had a deficiency, and 27.5% had a severe deficiency. Vitamin D levels were optimal in just 8.8% of the individuals.

The patients were stratified into groups according to serum 25(OH)D_3_ levels: severely deficient, deficient, insufficient, and optimal (Table 2). Free and bioavailable vitamin D levels were significantly higher (*p* < 0.001) in those with optimal 25(OH)D_3_ levels. Age (*p* = 0.028), ICU admission (*p* = 0.007), length of hospital stays (*p* = 0.018), Ranson score at admission (*p* = 0.045), and AP severity (*p* = 0.029) all showed significant group differences.

Severe AP was present only in the patients with severely deficient 25(OH)D_3_ levels. This group also had a higher proportion of cases with moderate acute pancreatitis (Figure 1). Moreover, seven patients were admitted to the ICU, and out of these, six had a severe deficiency of 25(OH)D_3_.

Table 2 displays the findings of the scoring system based on the degree of AP associated with vitamin D levels. Overall, 52 (65.0%) had mild AP, 19 (23.8%) had moderate AP, and 3 (3.8%) had severe AP, based on the revised Atlanta classification (Table 2).

According to computer tomography, 53 (66.3%) patients had interstitial and 22 (27.5%) patients had necrotizing AP. The latter was more common in cases with severely deficient levels of vitamin D (*p* = 0.013). In total, 19 (23.8%) of the patients had an alcoholic etiology, while 61 (76.2%) had another etiology (Figure 2).

AP did not exhibit a statistically significant difference between the groups based on stratified etiologic variables (*p* = 0.988). Because of the limited number of severe AP cases (n = 3), a new group, moderate-to-severe acute pancreatitis, was established with a sample size of 22 patients. This study had 80% power to detect medium effect sizes in differences between vitamin D categories. As such, some associations—particularly in logistic regression models—should be interpreted cautiously. In order to predict moderate-to-severe AP, an independent logistic regression was conducted. The results indicated the relationships between 25(OH)D_3_, free 25(OH)D_3_, bioavailable 25(OH)D_3_, necrotizing AP, alcoholic type of AP, and the CTSI score (Table 3). Moderate-to-severe AP was linked to lower levels of all three vitamin D variants. Conversely, more severe AP was associated with higher CT score values.

To improve robustness, a multivariate regression model was performed for variables that showed independent statistically significant correlations. The results showed that only necrotizing AP was statistically associated with moderate-to-severe AP (*p* = 0.017).

To independently predict ICU admission, a comparable study was carried out (Table 4). There were statistically significant associations between ICU admission and lower levels of 25(OH)D_3_, free 25(OH)D_3_, and bioavailable 25(OH)D_3_. Higher CTSI scores showed a similar trend. Admission to the intensive care unit was not linked to the type or cause of AP. The multivariate regression showed no significant correlations for ICU submission.

The impact of independent variables on the Ranson score and hospital stay duration was also examined using linear regression analysis. Table 5 and Table 6 present the findings. The patients who were older (*p* < 0.001) and had lower albumin levels (*p* = 0.018) were predicted to have higher Ranson scores.

There was no correlation between vitamin D levels and the Ranson score at admission. However, there was a negative correlation between the length of hospital stay and the levels of 25(OH)D_3_, free 25(OH)D_3_, and bioavailable 25(OH)D_3_ (Table 6).

The patients in the most significantly vitamin D inadequate group spent the longest time in the hospital (14.6 ± 15.2 days) (Table 2). Longer hospital stays were also associated with lower albumin levels. Longer hospital stays were also associated with the necrotizing form of AP and higher CTSI scores.

## 3. Discussion

In our study, the relationship between total, free, and bioavailable 25(OH)D_3_ and the severity of acute pancreatitis was evaluated. According to our findings, there is a direct correlation between patients who had lower levels of all three types of vitamin D, especially those who had a severe deficiency and more severe disease courses, such as necrotizing pancreatitis, longer hospital stays, and higher ICU admission rates.

Acute pancreatitis is still a challenging and potentially lethal disorder. At first, it originates locally in the pancreas and soon becomes a systemic disease. The systemic inflammatory response and risk of organ failure depend on many factors. Previous studies showed low vitamin D levels in acute pancreatitis [24,25]. Two studies also showed the relationship between 25(OH)D_3_ and the severity of AP [12,13]. Our research shows a significant correlation between more severe outcomes in acute pancreatitis and vitamin D deficiency, particularly in its severe form. Our study expands on earlier findings associating vitamin D insufficiency with poor outcomes in inflammatory disorders and is, to our knowledge, the first to objectively demonstrate the potential association between free and bioavailable 25(OH)D_3_ in AP severity. These results lend credence to the idea that vitamin D may influence the immunological and inflammatory reactions that contribute to the development of AP.

### Comparing Healthy Individuals

According to the research, vitamin D levels were much lower in AP patients than in the general healthy population [14,26]. Studies on healthy populations and recognized criteria indicate that blood concentrations of serum 25(OH)D_3_ should be between 75 and 125 nmol/L to be deemed optimal. Levels below 50 nmol/L indicate insufficiency, while concentrations below 30 nmol/L suggest severe deficiency [14,27]. Concentrations between 50 and 75 nmol/L are frequently categorized as insufficient. The average 25(OH)D_3_ concentration in our group of AP patients was 45.7 ± 25.3 nmol/L, which is significantly below the sufficiency level. As a result, most patients fell into the category of deficient or severely deficient.

In contrast to large-scale population-based studies, where severe deficiency is typically assumed to impact fewer than 10% of individuals, our AP patients showed a prevalence of severe deficiency (<30 nmol/L) of 27.5%, which is approximately three times higher [1]. Numerous disease-related factors, such as decreased levels of vitamin D-binding protein and albumin, altered vitamin D metabolism brought on by inflammation, and poor intestinal absorption, could be responsible for this disparity. These results confirm the idea that acute inflammatory conditions may worsen underlying insufficiency or quickly exhaust accessible vitamin D pools. They also highlight the increased susceptibility of AP patients to functional vitamin D deficiency.

Additionally, healthy individuals typically have higher levels of free and bioavailable vitamin D due to stable VDBP and albumin concentrations. However, AP patients often experience metabolic and inflammatory changes that may impact vitamin D absorption [11]. Because the levels of free and bioavailable vitamin D in our sample were significantly lower, it is possible that traditional measures of total vitamin D may underestimate functional insufficiency in AP. These findings support the hypothesis that, rather than being the sole outcome of population-wide trends, vitamin D deficiency in AP patients may be exacerbated by disease-specific factors, such as malabsorption, systemic inflammation, and altered protein synthesis [8,14].

Low levels of vitamin D have been shown to affect oxidative stress responses and autophagy, two processes that are essential for regulating inflammation in AP [28,29].

Numerous inflammatory conditions, such as sepsis, rheumatoid arthritis, and inflammatory bowel disease (IBD), have been linked to vitamin D deficiency [16,30,31,32,33,34,35]. Vitamin D-deficient patients with these illnesses have higher disease severity, more inflammation, and more extended hospital stays, just like those with AP. Similar to studies on AP, vitamin D deficiency is linked to increased intestinal permeability and bacterial translocation in IBD. Low vitamin D levels in sepsis are associated with immunological dysfunction and unfavorable outcomes, which supports vitamin D’s function as a vital regulator of inflammatory processes [35]. These similarities imply that vitamin D might modulate inflammatory illnesses in ways other than AP, which calls for more research.

Previous studies showed the correlation between serum vitamin D level and severity of AP and suggested that serum 25(OH)D_3_ level at admission could be a useful marker for predicting the severity of AP. As the bioavailability of vitamin D in different conditions could vary, our study showed that the serum level of vitamin D in patients with AP correlates with free 25(OH)D_3_ and bioavailable 25(OH)D_3_. This finding suggests that the detection of 25(OH)D_3_ at admission for predicting the severity of AP may be adequate for the initial stratification of AP severity. However, given the alterations in albumin and VDBP in AP, further studies are needed to clarify whether free or bioavailable vitamin D offers added clinical value.

Our research has several advantages. First of all, it is among the few studies that assesses both free and bioavailable vitamin D in addition to total serum vitamin D levels, which could provide a more accurate evaluation of vitamin D status in AP patients. Second, our findings are more reliable since we used a well-defined patient group with standardized diagnostic criteria, such as the Atlanta categorization. Third, there is substantial statistical support for the predictive utility of vitamin D status in AP severity thanks to the use of logistic regression analysis. However, our study has limitations as well. Our findings may be less generalizable because of the small sample size (N = 80). Furthermore, as this is an observational study, we cannot prove a link between the severity of AP and vitamin D deficiency. The absence of information on seasonal fluctuations in vitamin D levels, which may impact the prevalence of deficiency, is another drawback. Finally, even though our research points to a link between vitamin D levels and AP outcomes, further research is required to ascertain whether vitamin D treatment can enhance clinical results for AP patients. Despite these drawbacks, our results highlight the need for more studies in this field and add to the increasing body of data connecting vitamin D to inflammatory disorders. As a potential therapeutic intervention, future studies may explore rapid vitamin D repletion strategies, such as Stoss therapy, to assess whether vitamin D correction during AP can influence outcomes. Such interventions should be tested in randomized, controlled settings.

## 4. Materials and Methods

### 4.1. Patients

We included adult patients older than 18 who were hospitalized for a first episode of AP. Patients with previous episodes of AP were excluded. The diagnosis of AP was confirmed if two of the following three criteria were present: upper abdominal pain typical of AP, serum amylase and/or lipase elevated by at least 3× above the upper limit of normal, and radiological signs of AP. Patients were hospitalized in the gastroenterology department of internal medicine. We obtained a detailed patient history, determined etiological factors, and recorded co-pathology and regular therapy. The body mass index (BMI) was calculated as the weight in kilograms divided by square by square of height in meters. All patients received parenteral hydration from admission onwards and received a fluid bolus when there were signs of hypovolemia. Each patient enrolled in this study was started on 1.5 mL/kg/h of 0.9% saline solution with a prior bolus of 10 mL/kg if the patient was hypovolemic (moderate fluid resuscitation). The patients were monitored for fluid overload status, urine output, blood urea nitrogen levels, and blood pressure, and fluid resuscitation was adjusted accordingly. The fluid resuscitation approach was standard for all patients.

Peripheral blood was collected on admission for laboratory analysis: hemogram, electrolytes, calcium, albumin, urea, creatinine, glucose, lactate dehydrogenase, hepatogram, CRP, and vitamin D level. The vitamin D level was measured within 24 h of admission as serum 25-hydroxyvitamin D3 (25(OH)D_3_). Vitamin D deficiency was defined as a vitamin D level of 30 to 50 nmol/L, severe deficiency as below 30 nmol/L, and insufficient as 50 to 75. Normal values were defined as a level above 75 nmol/L.

The severity of AP was assessed according to the revised Atlanta 2012 criteria and classified as mild, moderate, or severe. Mild AP was defined by the absence of organ failure (OF) and local or systemic complications. Moderate acute pancreatitis was described as transient OF lasting less than 48 h and accompanied by local or systemic complications. Severe AP was defined as one or more organ failures with a duration of more than 48 h of OF. Endoscopic retrograde cholangiopancreatography was performed when biliary pancreatitis with signs of biliary obstruction was diagnosed. Because of the need for scoring (Ranson score, CTSI, Revised Atlanta classification) and monitoring of the clinical picture, peripheral blood was collected several times.

All patients underwent abdominal ultrasound, and some patients (especially those with suspected severe or necrotizing pancreatitis) underwent contrast-enhanced CT of the abdominal organs.

The Ethics Committee approved the study protocol for Human Research of the Medical Ethics Commission of the Republic of Slovenia. All patients gave written informed consent for this study before inclusion.

### 4.2. Blood Analyses

Blood was collected into a biochemical blood tube (4 mL). Albumin and total 25(OH)D_3_ were analyzed from fresh serum after blood collection, while the aliquot for vitamin D binding protein (DBP) was stored at minus 80 degrees Celsius until analysis. All samples were analyzed simultaneously. Measurements were performed at the Clinical Institute of Clinical Chemistry and Biochemistry (University Medical Centre, Ljubljana).

25(OH)D_3_, S-albumin, and S-DBP in serum were measured in all the participants using the following methods: The concentration of 25(OH)D_3_ vitamin was measured using a competitive luminescent immunoassay with intra-laboratory CV < 6% and a limit of quantification of 6 nmol/L (Architect analyzer, Abbott Diagnostics, Lake Forest, IL, USA), the ADVIA^®^ 1650 Chemistry Albumin BCP Assay (Siemens, New York, NY, USA), and Human Vitamin D Binding Protein was measured with ELISA (MyBioSource, Inc., San Diego, CA, USA); the limit of quantification was 31 mg/L. Free and bioavailable 25(OH)D3 were calculated using Bikle’s adaptation of the modified Vermeulen equation, based on total 25(OH)D_3_, VDBP, and albumin levels [36]. The formula is freely available online [https://vidvicic.com/dcalc/ (accessed on 16 November 2024)] and has been validated for use in inflammatory conditions.

### 4.3. Statistical Analysis

Statistical analyses were performed using SPSS 22.0 software (SPSS Inc, Chicago, IL, USA). Continuous variables were presented as mean values with standard deviation, and categorical variables were presented as frequencies and percentages. The normality of the distribution was confirmed using the Shapiro–Wilk test. The severity of acute pancreatitis, namely, the prediction of moderate-to-severe AP according to the Atlanta classification, was considered as a dependent variable, as only 3 cases were registered as severe. The remaining variables were categorized as independent. The main investigated independent variables in the statistical analysis were the levels of total 25(OH)D_3_, free 25(OH)D_3_, and bioavailable 25(OH)D_3_ if they could predict the severity of AP. The Endocrine Society cut-off values were used to assess total serum 25(OH)D_3_ level target concentrations for the optimal vitamin D effect, i.e., 75–125 nmol/L, insufficiency, i.e., 50–75 nmol/L, and deficiency, i.e., <50 nmol/L [14].

The main differences in the dependent variables were analyzed using a One-way ANOVA test for those variables that were normally distributed. In the case of non-parametric non-normally distributed variables, we used the Kruskal–Wallis test for comparisons. Pearson’s chi-squared test was used to compare categorical variables. The correlations between dependent and independent variables were measured by Spearman’s correlation test. Logistic and multivariate logistic regressions with odds ratios were used to predict moderate-to-severe acute pancreatitis. All tests were considered as statistically significant at *p* < 0.05.

## 5. Conclusions

Acute pancreatitis (AP) patients are very likely to have deficiency or severe deficiency of vitamin D. According to the revised Atlanta classification, higher severity, longer hospital stays, and higher ICU admission rates are all indicative of more severe disease, which is substantially correlated with lower concentrations of total 25(OH)D_3_ and its free and bioavailable forms. Specifically, moderate-to-severe AP and necrotizing pancreatitis are highly associated with severe vitamin D deficiency and are both connected to worse clinical outcomes.

Low levels of each of the three types of vitamin D are correlated to disease severity, as confirmed by multivariate logistic regression analysis. In order to further stratify risk in AP patients, these results demonstrate the therapeutic significance of measuring not only total vitamin D but also free and bioavailable vitamin D. Mechanistically, immunological dysregulation, increased intestinal permeability, and disruptions in calcium homeostasis are among the ways that vitamin D deficiency may advance the course of disease.

Similar immunopathological pathways are seen when compared to other inflammatory illnesses, such as sepsis and inflammatory bowel disease, which supports the findings’ biological plausibility. All of these data point to the necessity of additional interventional research to ascertain whether vitamin D administration could enhance the prognosis of patients suffering from acute pancreatitis.

## Figures and Tables

**Figure 1 ijms-26-05695-f001:**
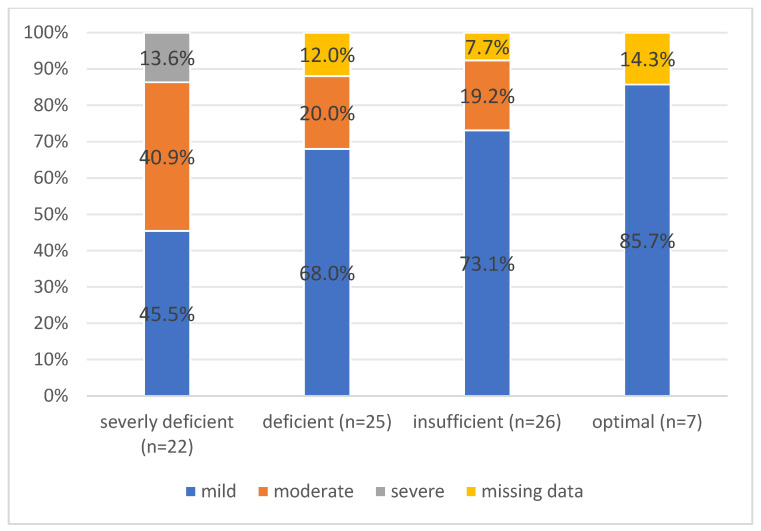
The proportion of patients with mild, moderate, and severe acute pancreatitis according to the groups of vitamin D deficiency.

**Figure 2 ijms-26-05695-f002:**
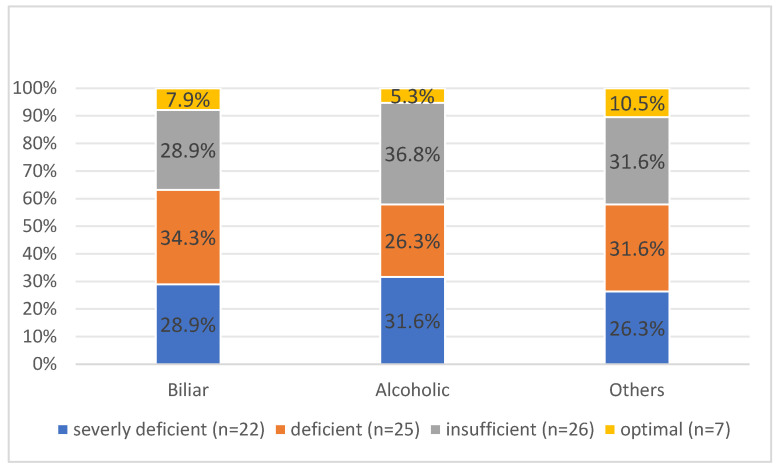
The etiology of AP across vitamin D level groups.

**Table 1 ijms-26-05695-t001:** Basic characteristics of enrolled patients with AP.

	All Patients (N = 80)
Age [years]	58.8 ± 16.1
Gender	
Male/female	45/35
BMI	28.3 ± 5.4
VDBP [mg/mL]	146.5 ± 82.6
25(OH)D_3_ [nmol/L]	45.7 ± 25.3
Vit D deficiency	
Severe deficient < 30 nmol/L	22 (27.5%)
Deficient 30–50 nmol/L	25 (31.3%)
Insufficient 50–75 nmol/L	26 (32.5%)
Optimal > 75 nmol/L	7 (8.8%)
Free 25(OH)D_3_	28.5 ± 22.8
Bioavailable 25(OH)D_3_	9.3 ± 7.8
Supplementation of vitamin D	28 (35.0%)
Etiology	
Biliar	38 (47.5%)
Alcoholic	19 (23.8%)
Hyperlipemic	4 (5.0%)
Other	15 (18.8%)
Missing data	4 (5.0%)
Smoking	30 (37.5%)
Number of packs/year	11.7 ± 16.2
Alcohol consumption	
No	29 (36.3%)
1 to 7×/week (low)	22 (27.5%)
8–14×/week (moderate)	7 (8.6%)
14–20/week (high)	9 (11.3%)
>20×/week (severe)	4 (5.0%)
Missing data	9 (11.3%)

**Table 2 ijms-26-05695-t002:** Comparison of patients stratified into groups according to 25(OH)D_3_ levels.

	Vit D Deficiency				
	Severely Deficient (n = 22)	Deficient (n = 25)	Insufficient (n = 26)	Optimal (n = 7)	*p*-Value
Age	53.8 ± 14.0	55.6 ± 15.0	62.9 ± 17.1	71.2 ± 15.2	0.028
Gender					
Male/female	12/10	16/9	13/13	4/3	0.789
25(OH)D_3_ [nmol/L]	19.0 ± 5.1	38.0 ± 6.4	61.3 ± 6.3	99.3 ± 26.9	<0.001
Free 25(OH)D_3_ [pmol/L]	9.9 ± 4.4	24.5 ± 16.3	36.9 ± 17.3	69.7 ± 28.9	<0.001
Bioavailable 25(OH)D_3_ [nmol/L]	3.1 ± 1.5	7.9 ± 5.1	11.8 ± 5.6	25.0 ± 9.8	<0.001
Pts with Ranson after 48 h ≥3	0	1 (4.0%)	0	0	0.509
CRP [mg/L]	80.7 ± 123.4	50.6 ± 62.5	54.5 ± 70.3	28.0 ± 49.0	0.493
Admission to ICU	6 (27.3%)	1 (4.0%)	0	0	0.007
Hospitalization duration [days]	14.6 ± 15.2	9.7 ± 9.3	6.9 ± 3.8	8.0 ± 5.5	0.018
**Type of AP**					
Class of AP (Atlanta)					0.029
Mild	10 (45.5%)	17 (68.0%)	19 (73.1%)	6 (85.7%)	
Moderate	9 (40.9%)	5 (20.0%)	5 (19.2%)	0	
Severe	3 (13.6%)	0	0	0	
Missing data	0	3 (12.0%)	2 (7.7%)	1 (14.3%)	
CT finding of AP					0.013
Interstitial	10 (45.5%)	18 (72.0%)	19 (73.1%)	6 (85.7%)	
Necrotizing	12 (54.5%)	5 (20.0%)	5 (19.2%)	0	
Missing data	0	2 (8.0%)	2 (7.7%)	1 (14.3%)	
CT score; CTSI	2.7 ± 3.2	1.8 ± 3.2	1.5 ± 3.2	0	0.191
CTSI > 3	10 (45.5%)	6 (24.0%)	4 (15.4%)	0	0.087
Etiology of AP					0.988
Biliar	11 (50.0%)	13 (52.0%)	11 (42.3%)	3 (42.9%)	
Alcoholic	6 (27.3%)	5 (20.0%)	7 (26.9%)	1 (14.3%)	
Other	5 (22.7%)	6 (24.0%)	6 (23.1%)	2 (28.6%)	

**Table 3 ijms-26-05695-t003:** Logistic regression prediction with independent factors for moderate-to-severe AP.

	B	OR	95% CI	*p*-Value
Gender male	0.370	1.448	0.521–4.030	0.478
Age [years]	−0.02	0.98	0.95–1.01	0.266
BMI	−0.07	0.93	0.84–1.03	0.167
25(OH)D_3_ [nmol/L]	−0.04	0.96	0.93–0.99	0.006
Free 25(OH)D_3_ [pmol/L]	−0.07	0.93	0.89–0.98	0.003
Bioavailable 25(OH)D_3_ [nmol/L]	−0.24	0.79	0.67–0.92	0.002
Albumin	−0.106	0.899	0.807–1.002	0.055
Etiology—alcoholic	1.405	4.074	1.351–12.286	0.013
CT—Necrotizing	6.996	1092	65.236–18,279.158	<0.001
CTSI score	0.791	2.206	1.581–3.077	<0.001

**Table 4 ijms-26-05695-t004:** Logistic regression prediction with independent factors for admission to the ICU.

	B	OR	95% CI	*p*-Value
Gender male	0.680	1.974	0.358–10.893	0.435
Age [years]	−0.02	0.98	0.94–1.03	0.468
BMI	0.03	1.03	0.89–1.19	0.712
25(OH)D_3_ [nmol/L]	−0.14	0.87	0.78–0.97	0.014
Free 25(OH)D_3_ [pmol/L]	−0.39	0.68	0.47–0.97	0.036
Bioavailable 25(OH)D_3_ [nmol/L]	−1.04	0.35	0.13–0.98	0.046
Albumin	−0.061	0.941	0.801–1.104	0.456
Etiology—alcoholic	0.891	2.437	0.493–12.053	0.275
CT—Necrotizing	20.441	753,888	0.000	0.997
CTSI score	0.541	1.718	1.224–2.411	0.002

**Table 5 ijms-26-05695-t005:** Linear regression prediction with independent factors for the Ranson score at admission.

	Unstandardized B	Standardized Beta Correlation	t	95% CI for B	*p*-Value
Gender, male	−0.257	−0.132	−1.169	−0.695–0.181	0.246
Age	0.033	0.551	5.792	0.022–0.045	<0.001
BMI	−0.019	−0.108	−0.921	−0.061–0.022	0.360
25(OH)D_3_ [nmol/L]	0.01	0.08	0.70	0.01–0.01	0.485
Free 25(OH)D_3_ [pmol/L]	0.01	0.07	0.62	−0.01–0.01	0.540
Bioavailable 25(OH)D_3_ [nmol/L]	0.01	0.03	0.24	0.03–0.03	0.808
Albumin	−0.051	−0.265	−2.414	−0.092–(−0.009)	0.018
Etiology—alcoholic	−0.193	−0.084	−0.724	−0.724–0.338	0.471
CT—Necrotizing	0.215	0.103	0.875	−0.275–0.705	0.384
CTSI score	0.016	0.052	0.433	−0.058–0.091	0.666

**Table 6 ijms-26-05695-t006:** Linear regression prediction with independent factors for hospitalization duration.

	Unstandardized B	Standardized Beta Correlation	t	95% CI for B	*p*-Value
Gender, male	1.639	0.079	0.675	−3.202–6.479	0.502
Age	0.038	0.059	0.506	−0.111–0.187	0.614
BMI	−0.401	−0.201	−1.733	−0.862–0.060	0.087
25(OH)D_3_ [nmol/L]	−0.11	−0.27	−2.42	0.20–(−0.02)	0.018
Free 25(OH)D_3_ [pmol/L]	−0.11	−0.24	−2.08	−0.22–(−0.01)	0.041
Bioavailable 25(OH)D_3_ [nmol/L]	−0.35	−0.25	−2.19	−0.66–(−0.03)	0.032
Albumin	−0.498	−0.244	−2.154	−0.959–(−0.037)	0.035
Etiology—alcoholic	4.249	−0.176	0.524	−1.308–9.805	0.132
CT—Necrotizing	11.599	0.503	4.941	6.920–16.279	<0.001
CTSI score	1.569	0.459	4.324	0.845–2.293	<0.001

## Data Availability

The raw data supporting the conclusions of this article will be made available by the corresponding author on request.

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
