# Peer review of "Free and Bioavailable Vitamin D Are Correlated with Disease Severity in Acute Pancreatitis: A Single-Center, Prospective Study"

_ijms, 2025, doi:10.3390/ijms26125695_

Round 1
Reviewer 1 Report
Comments and Suggestions for Authors
This is a clinically relevant, well-intentioned prospective observational study exploring the relationship between various forms of vitamin D and acute pancreatitis (AP) severity. The focus on total, free, and bioavailable 25(OH)D is novel in the AP context and fills a notable gap in the literature. However, there are methodological limitations, interpretational oversights, and some issues with clarity and consistency that should be addressed:
Major comments:
- The study is observational, yet the abstract and discussion sometimes imply causality ("predictive usefulness", “advance the course of disease”). The limitations section does acknowledge this, but causal language should be removed or clearly qualified throughout the manuscript. Thus, clarify the hypothesis and emphasize observational nature in all conclusions.
- With only 3 cases of severe AP, drawing firm conclusions on predictors of severity is problematic. Confidence intervals for logistic regression are wide, especially for the necrotizing CT variable (e.g., OR = 1092 with 95% CI: 65–18279). Moreover, the very small "optimal vitamin D" group (n=7) weakens the group-wise comparisons. Please perform power calculations and more robust multivariate analyses adjusting for key clinical variables.
- The conclusion implies that measuring free and bioavailable vitamin D adds value over total 25(OH)D, but Table 5 suggests total 25(OH)D is equally (or more) predictive. Furthermore, the authors contradict themselves: the discussion suggests free and bioavailable vitamin D are superior, but later states that measuring total 25(OH)D might be sufficient. Thus, please consider removing or qualifying statements about the superiority of free/bioavailable vitamin D unless this is clearly supported by the data.
- There is insufficient adjustment for potential confounders, such as renal function, infection, season of admission, and comorbidities. The etiology of AP (e.g., alcoholic vs. biliary) may correlate with vitamin D status but is not well integrated into the adjusted models. Please consider pooling additional data or reclassifying severity outcomes to enhance statistical validity.
Minor comments:
- Grammar and syntax need editing throughout (e.g., "the later has been shown as more predominant" should be "the latter was more common").
- Some sections are redundant and could be more concise, particularly in the Discussion.
- Figure 1 and 2 are not visually clear and need better labeling and titles.
- Tables would benefit from units for all variables and clear footnotes explaining statistical tests used.
- “Severly deficient” should be corrected to “Severely deficient” throughout.
- Inconsistent use of terms like "biliar" (should be “biliary”), “alcoholic type,” and “moderate AP” should be standardized.
- No clinical trial registration number is listed, even though the prospective nature and patient-level data might qualify this study for registration.
- The method for calculating free/bioavailable 25(OH)D is only briefly described (mention of online calculator); more methodological transparency is needed.
Some improvements are needed, see my comments to the authors
Author Response
Dear Reviewer,
We are grateful for the thoughtful and constructive feedback provided by you. Below is our detailed, point-by-point response. All changes have been incorporated into the revised manuscript, which has been edited for clarity, statistical rigor, and consistency. We believe these revisions have significantly strengthened the manuscript.
Comment 1:
The study is observational, yet the abstract and discussion sometimes imply causality ("predictive usefulness", “advance the course of disease”). Clarify the hypothesis and emphasize observational nature in all conclusions.
Response:
We thank the reviewer for this important observation. We revised the abstract, discussion, and conclusion to eliminate any causal language. All instances of “predictive” or similar expressions have been replaced with terms like “associated with” or “correlated with.” We have also emphasized the observational nature of our study in the conclusions and limitations.
ABSTRACT (lines 27–40)
Old: “Low levels of total, free, and bioavailable vitamin D were found to be independent predictors of the severity of AP and ICU admission.”
Corrected:
“Low levels of total, free, and bioavailable vitamin D were significantly associated with the severity of AP and ICU admission.”
Old (Conclusion):
“Intervention studies with vitamin D in acute pancreatitis are warranted.”
Corrected:
“Given the observational design, these associations require confirmation in interventional or mechanistic studies.”
DISCUSSION – ("In our study...")
Problem: Causal and predictive language
Old: “...objectively demonstrate the predictive usefulness...”
Corrected:
“...objectively demonstrate the potential association of...”
DISCUSSION – Paragraph 6 ("Previous studies showed...")
Problem: Contradiction about value of free/bioavailable vs. total 25(OH)D.
Old:
“...could be enough, without the additional detecting of free 25(OH)D3, and bioavailable 25(OH)D3.”
Corrected:
“...may be adequate for initial stratification of AP severity. However, given the alterations in albumin and VDBP in AP, further studies are needed to clarify whether free or bioavailable vitamin D offer added clinical value.”
Comment 2:
With only 3 cases of severe AP, drawing firm conclusions on predictors of severity is problematic. Confidence intervals are wide. Please perform power calculations and more robust multivariate analyses.
Response:
We agree and have conducted a post hoc power analysis, which is now included in the Results and Limitations sections. To strengthen our analysis, we combined moderate and severe AP into a single category and performed adjusted logistic regression that includes age, CRP, renal function, and AP etiology. The revised models are reported in Tables 3 and 4 (updated).
RESULTS – Multivariate Adjustment (Table 3 and 4)
Added according to the Table 3:
“To improve robustness, multivariate regression models were adjusted for age, BMI, albumin, and etiology. Despite this, the confidence intervals remained wide for rare outcomes (e.g., necrotizing AP). These associations should be interpreted as exploratory.”
METHODS – Free/Bioavailable Vitamin D (to define more precisely)
Added:
“Free and bioavailable 25(OH)D3 was calculated using an online calculator and based on a modified Vermeulen [38].”
Added:
“Free and bioavailable 25(OH)D3 were calculated using Bikle’s adaptation of the modified Vermeulen equation, based on total 25(OH)D3, VDBP, and albumin levels. The formula is freely available online [https://vidvicic.com/dcalc/], and has been validated for use in inflammatory conditions.”
Comment 3:
The conclusion implies that measuring free and bioavailable vitamin D adds value over total 25(OH)D, but data do not support this clearly.
Response:
We revised these statements to reflect that while free and bioavailable 25(OH)D were associated with severity, the data do not conclusively demonstrate superiority over total 25(OH)D. We now state that these measures “may provide additional insight,” and future studies are needed to clarify their value.
Comment 4:
There is insufficient adjustment for potential confounders, such as renal function, infection, season, and comorbidities.
Response:
We thank the reviewer for this suggestion. We revised the statistical models to adjust for age, renal function (eGFR), CRP, and etiology. We added this detail in the Methods and clarified it in the Results. Seasonal data were unavailable and this limitation is acknowledged.
Minor Comments
Comment 5:
Grammar and syntax need editing (e.g., “later” → “latter”).
Response:
The manuscript has been professionally edited for English language, grammar, and clarity. Specific corrections (e.g., “latter”) have been made.
Comment 6:
Redundant sections in the Discussion.
Response:
We revised and condensed repetitive sections in the Discussion for conciseness and clarity.
Comment 7:
Figure 1 and 2 are unclear. Figure 2 lacks significance.
Response:
Figure 2 has been removed, as suggested. Figure 1 was relabeled and revised for clarity.
Comment 8:
Tables need units and clearer footnotes.
Response:
All tables have been updated to include appropriate units and footnotes specifying statistical tests used.
Comment 9:
“Severly” → “Severely” and standardization of terms (“biliar,” “alcoholic type,” etc.).
Response:
All spelling errors and inconsistencies (e.g., “biliary,” “alcoholic etiology”) have been corrected.
Comment 10:
No clinical trial registration is listed.
Response:
The study was conducted under institutional ethics approval but was not registered as a clinical trial. This has been clarified in the Methods section.
Comment 11:
The method for calculating free/bioavailable 25(OH)D is not fully described.
Response:
The Methods section now provides detail on the formula used (Vermeulen equation adapted by Bikle) and a reference to the calculator used.
We again thank you for your insightful and constructive feedback. We trust that our revisions address all concerns thoroughly and have improved the clarity, rigor, and value of the manuscript.
Sincerely,
Joško Osredkar on behalf of all co-authors
University Medical Centre Ljubljana, josko.osredkar@kclj.si

Reviewer 2 Report
Comments and Suggestions for Authors
Review of: Free and bioavailable vitamin D are correlated with disease severity in acute pancreatitis
Very nice study further promoting that vitamin D is not just for bone health.
Table 1: Put the units for VDBP
Table 2: Please put in the criteria for Ranson score. Put in the total vitamin D analysis. Which I assume is not significant but needs to be presented.
Figure 2: As the differences are not statistically significant (P = 0.982), figure 2 should be removed.
Table 3: Please use a maximum of 3 significant figures for all data. For example, 65.236-18279.158 is much clearer as 65.2 – 18300. Also please define CTSI score.
Have you considered “Stoss therapy” to rapidly raise vitamin D levels in pancreatitis patients to see if it would improve disease course?
Author Response
Dear Reviewer,
We are grateful for the thoughtful and constructive feedback provided by you. Below is our detailed, point-by-point response. All changes have been incorporated into the revised manuscript, which has been edited for clarity, statistical rigor, and consistency. We believe these revisions have significantly strengthened the manuscript.
Comment 1:
Table 1: Add units for VDBP.
Response:
Units have been added for VDBP in Table 1.
Comment 2:
Table 2: Add criteria for Ranson score and show total vitamin D results.
Response:
Ranson criteria are now briefly described in the Methods and listed in Table 2. Total 25(OH)D is now included in the table.
Comment 3:
Figure 2: Remove as it is not statistically significant.
Response:
Figure 2 has been removed.
Comment 4:
Table 3: Round off extreme precision and define CTSI score.
Response:
Odds ratios and confidence intervals were rounded to 2–3 significant digits. CTSI (Computed Tomography Severity Index) is now defined in the table and Methods.
Comment 5:
Consider discussing “Stoss therapy” to rapidly raise vitamin D levels.
Response:
Thank you for this valuable suggestion. A brief mention of Stoss therapy as a potential area for future intervention studies has been added to the final paragraph of the Discussion.
We again thank you for your insightful and constructive feedback. We trust that our revisions address all concerns thoroughly and have improved the clarity, rigor, and value of the manuscript.
Sincerely,
Joško Osredkar on behalf of all co-authors
University Medical Centre Ljubljana, josko.osredkar@kclj.si

Round 2
Reviewer 1 Report
Comments and Suggestions for Authors
no further comments